# RTX Toxins of Animal Pathogens and Their Role as Antigens in Vaccines and Diagnostics

**DOI:** 10.3390/toxins11120719

**Published:** 2019-12-10

**Authors:** Joachim Frey

**Affiliations:** Vetsuisse Facutly, University of Bern, 3012 Bern, Switzerland; joachim.frey@vetsuisse.unibe.ch

**Keywords:** cytotoxicity, host specificity, RTX receptors, diagnostic applications, vaccines, disease resistance

## Abstract

Exotoxins play a central role in the pathologies caused by most major bacterial animal pathogens. The large variety of vertebrate and invertebrate hosts in the animal kingdom is reflected by a large variety of bacterial pathogens and toxins. The group of repeats in the structural toxin (RTX) toxins is particularly abundant among bacterial pathogens of animals. Many of these toxins are described as hemolysins due to their capacity to lyse erythrocytes in vitro. Hemolysis by RTX toxins is due to the formation of cation-selective pores in the cell membrane and serves as an important marker for virulence in bacterial diagnostics. However, their physiologic relevant targets are leukocytes expressing β2 integrins, which act as specific receptors for RTX toxins. For various RTX toxins, the binding to the CD18 moiety of β_2_ integrins has been shown to be host specific, reflecting the molecular basis of the host range of RTX toxins expressed by bacterial pathogens. Due to the key role of RTX toxins in the pathogenesis of many bacteria, antibodies directed against specific RTX toxins protect against disease, hence, making RTX toxins valuable targets in vaccine research and development. Due to their specificity, several structural genes encoding for RTX toxins have proven to be essential in modern diagnostic applications in veterinary medicine.

## 1. Introduction

RTX (repeats in the structural toxin) are large pore-forming proteins named due to the presence of characteristic arrays of glycine and aspartate-rich nonapeptide repeats in the C-terminal half of their amino acid sequence. They constitute a subfamily of α-pore-forming protein toxins [1]. The repeats contain the common sequence structure G–G–X–G–(N/D) –D–X–(L/I/V/W/Y/F)–X (where X can be any amino acid) and are present in variable numbers ranging from six to more than 50 copies. These nonapeptide repeats form a parallel β-roll motif within a right-handed spiral, which binds Ca^2+^ ions held by two sterically neighboring nonapeptide repeats [2,3,4]. Several RTX toxins have been shown to bind Ca^2+^ in solution and to be dependent on Ca^2+^ to exhibit their pore-forming and cytolytic activity. The prototype RTX toxin is the α-hemolysin HlyA identified in human and animal pathogenic *Escherichia coli.* This protein was initially detected, like most other RTX toxins, by the characteristic hemolytic halo surrounding bacterial colonies grown on agar medium containing sheep erythrocytes [5,6]. This characteristic is still used in diagnostics as a first rapid differentiation of potentially pathogenic variants of a given species. In addition to binding Ca^2+^, the pore-forming and lytic activity of RTX toxins require post-translational modification by an activating acyltransferase. The RTX toxin (generally a large protein of >100 kDa molecular mass, abbreviated RtxA) is synthesized in the form of an inactive protoxin and is post-translationally modified by the activator acyltransferase activator (RtxC) catalyzing the attachment of two fatty acids to conserved lysine residues located upstream of the glycine-rich repeats [7,8,9] (Figure 1). Furthermore, the structure of RTX toxins is characterized by strong hydrophobic domains upstream of the acylated lysine residues and an amphipathic helix at the N-terminal moiety of the protein, both involved in pore formation. While many RTX toxins show this basic structure, a second group, composed of very large multifunctional RTX toxins have also been discovered. These proteins, which have molecular masses up to 360 kDa, are named multifunctional autoprocessing repeats-in-toxin (MARTX) toxins and are found in both human and animal pathogens [10] (Figure 1). They contain additional functional domains, mostly upstream of the basic RTX structures.

A prominent characteristic of RTX proteins is that they are secreted by a committed type I secretion system (T1SS), which constitutes one of the major export systems used by Gram-negative bacteria to secrete proteins into their external medium. T1SS allows efficient secretion of the structural RTX proteins across the inner and outer membrane. Genes encoding the T1SS proteins RtxB and RtxD are generally located together with the activator and structural RTX genes on a polycistronic operon and are tightly regulated. They form, with the intrinsic outer membrane protein TolC, the secretion channel (Figure 1). Secretion is initiated by recognition of a non-cleavable secretion signal at the C-terminus of the RTX protein, and the export occurs without periplasmic intermediates [8,13,14,15,16]. The tight link between the specific domains of the RTX toxins and their posttranslational activation with their proper T1SS, reveals the intrinsic need of RTX toxins to be secreted efficiently as their properties would be deleterious when kept in the cytoplasm of bacteria [17,18,19].

The basic genetic structure of RTX toxins is distinctive and includes the structural gene “A”, the activator gene “C” and T1SS genes “B” and “D”. The archetypical operon for RTX toxins is organized in the form of a polycistronic operon, p*CA*‖*BD*, where p is the promoter and ǁ a transcription attenuator, which reduces the expression of the T1SS genes B and D, which are integral membrane proteins that are needed at lower amounts than the RTX toxins themselves [7,8,14,15,20]. Gene *tolC*, is also required but not exclusive for the T1SS. It is generally not part of the operon and is instead located at a distant site on the bacterial chromosome [13] (Figure 1). However, other genetic structures are known, in particular for composite RTX toxins, which also exhibit beside the core toxin, other biochemical functions (Figure 1).

The most evident effect of RTX toxins on their target cells is their cytolytic and hemolytic activities. The latter is directly visible by the hemolytic halo surrounding bacterial colonies grown on blood agar medium. The pore-forming activity causing the lytic process requires the binding of Ca^2+^ to the glycine-rich repeats and acylation [21]. Although most RTX toxins have a pronounced hemolytic activity in vitro, leukocytes have been determined to be their main targets. At sublytic concentrations, RTX toxins bind to β_2_-integrins and induce signaling cascades leading to apoptosis that results in inflammation, tissue lesions, and, finally, to disease [22,23,24,25,26,27]. Initial contact and binding of RTX proteins with their targets were shown to require acylation of two specific lysine sites that leads to a conformational stabilization of the protein and Ca^2+^ binding at the glycine-rich nonapeptide repeats [28,29,30,31].

The N-terminal domain that contains an amphipathic segment and several hydrophobic domains (Figure 1) is essential for the hemolytic and cytotoxic activity of RTX toxins. These hydrophobic domains can form α-helical structures that insert into lipid bilayers and form pores, which are finally responsible for membrane permeabilization [4,32,33,34].

While several fundamental mechanisms of RTX toxins have been analyzed in human pathogens, knowledge of RTX toxins from bacteria causing animal diseases predominantly has led to an understanding of the causes and spread of epizootic diseases and the development of diagnostic tests and vaccines. This review provides an overview of the RTX toxins that play an important role in the detection and characterization of animal pathogens and the development of potent therapeutic and preventive medicines. The impact of RTX toxins from important animal pathogens is discussed in terms of basic knowledge of the molecular mechanisms of pathogenicity, their use for development of diagnostic tools, and their role in the design and development of potential vaccines against major animal epizootics caused by RTX-producing bacterial pathogens.

## 2. RTX Toxins of Animal Pathogens

### 2.1. LktA Leukotoxin of Mannheimia (Pasteurella) Haemolytica

The leukotoxin (LktA) of *Mannheimia* (*Pasteurella*) *haemolytica* is an archetypical RTX leukotoxin and is currently the most studied member of the RTX toxins of *Pasteurellaceae*. *M. haemolytica* is the causative agent of mannheimiosis (previously known as pneumonic pasteurellosis), an important respiratory disease complex that leads to severe morbidity and mortality of cattle, sheep, and goats. LktA, a pore-forming, weakly hemolytic RTX toxin appears to be the main virulence factor of *M. haemolytica* [25,35,36,37,38]; however, other *Pasteurellaceae* such as *Bibersteinia trehalosi* also secrete LktA. A deletion in the *lktA* leukotoxin gene of *M. haemolytica* results in a strain that is unable to cause pulmonary lesions, in spite of its capacity to colonize the upper respiratory tract of calves, showing the important role of LktA in virulence [39]. Furthermore, necrosis of ruminant neutrophils can be induced in vitro with purified leukotoxin, indicating the cytotoxic effect of LktA during infections with *M. haemolytica* [40,41]. LktA is only cytotoxic for ruminant leukocytes, revealing the role of this toxin in the specific host range of *M. haemolytica,* which is limited to ruminants [42,43,44]. While the overall structure of leukotoxin is well conserved, studies of leukotoxin genes of several bovine and ovine *M. haemolytica* isolates revealed a complex mosaic structure indicating that it has evolved from a series of inter- and intraspecies horizontal gene transfers between the major lineages of *M. haemolytica* [45,46,47,48,49]. These variations of structure, seen in different subtypes of LktA, have an impact on the cytotoxic potency but seem to not have an effect on the host specificity for ruminant leukocytes [46]. Host specificity of LktA results from particular interactions with the β subunit CD18 of the β_2_ integrins of bovine, ovine, and caprine leukocytes [50,51,52]. CD18 is conserved among these ruminants and distinct from CD18 of other species [50,51,53,54]. In particular, the integrin-EGF-3 domain of CD18 has been shown to be critical for LktA species-specific susceptibility [55]. In a study using murine P815 mastocytoma cells that were stably transfected with the ovine CD18 gene, the mastocytoma cells became susceptible to LktA, while untransfected cells were not [51]. Another study showed that a human β_2_ integrin deficient leukocyte cell line that was transfected with the genes for the bovine α_L_β_2_ integrin (CD11a_bovine_/CD18_bovine_) was susceptible to the cytotoxic effect of LktA, while the same cell line, when transfected with the CD11a_bovine_/CD18_human_ genes to produce a hybrid bovine/human α_L_β_2_ integrin, was resistant to LktA [56]. Thus, LktA of *M. haemolytica* exerts its cytotoxicity to ruminant leukocytes via a specific interaction with a CD18 moiety conferring a species-specific pathogenic phenotype to *M. haemolytica*. The cytotoxic potency appears to depend on the modular composition of a given LktA subtype and the quantity that is produced, as well as on the CD18 allele present in the affected animal.

A recent study analyzing missense variants in the integrin β_2_ gene (*ITGB2*) encoding CD18 of more than 1200 cattle from 50 different breeds revealed that missense variants in the CD18 signal peptide affect LktA binding. Consequently, animals carrying certain alleles can be more susceptible to the effects of LktA, and hence more prone to mannheimiosis [57]. These findings could be important to cattle breeding since *M. haemolytica* is spread widely in cattle herds. The results further identified a potent class of LktA inhibitors that potentially protect cattle from cytotoxic effects of LktA during acute lung infections and may result in novel drugs for non-antibiotic therapy of mannhemiosis.

Czuprinski and collaborators have shown that LktA induces apoptosis in bovine lymphoblastoid cells (BL-3 cells) in a caspase-9-dependent manner by disrupting the outer mitochondrial membrane. This causes the collapse of the mitochondrial membrane potential and release of cytochrome C. They demonstrated that this process occurred through internalization of LktA and binding to the mitochondrial matrix protein cyclophilin D, assumedly by dynamin-2-triggered targeting [27,58,59]. Targeting LktA to the mitochondrial membrane of BL-3 cells requires the amino-terminal 31 amino acids of LktA [60]. It is interesting to note that mitochondrial targeting motifs were found in the first 54 amino-terminal residues in several RTX toxins, including LktA, ApxII, and the enterohemorrhagic *E. coli* hemolysin EhxA, but not in ApxI, ApxIII, the *E. coli* hemolysin HlyA, and the leukotoxin LtxA of the human pathogen *Aggregibacter actinomycetemcomitans* [60]. This amino terminus of RTX toxins that forms an amphipathic helix (Figure 1), is the most divergent domain, suggesting variation in their capacity to interact with specific lipid membranes in their target cells. Hence, the initial binding of RTX toxins to CD18 as well as targeting the toxin to its final destination seems to be specified by different domains.

Immunological and serological analyses of cattle suffering from *M. haemolytica* infections revealed a strong induction of immune response to LktA in diseased animals but not in clinically asymptomatic carriers. Furthermore, experimental addition of recombinant LktA to standard vaccines against mannheimiosis augmented the vaccine efficacy. This led to the development of current commercial vaccines such as OneShot® (Zoetis, Parsippan, NJ, USA), which are primarily composed of toxoided culture supernatant (containing significant amounts of LktA) and inactivated bacteria. Furthermore, live attenuated LktA-secreting bacteria have been investigated as potential vaccines that are effective not only against LktA-producing *M. haemolytica*, but also against other LktA-producing *Pasteurellaceae* such as *B. trehalosi* (Table 1) [61,62].

### 2.2. ApxI, ApxII, ApxIII, and ApxIV from Actinobacillus Pleuropneumoniae, a Multi-RTX Toxin-Producing Pathogen Causing Porcine Pleuropneumonia

*Actinobacillus pleuropneumoniae* is the etiological agent of porcine pleuropneumonia, a severe and highly contagious acute or chronic infectious disease. It has caused severe losses in the swine industry worldwide. In China, the biggest pig producing country, yearly losses of porcine pleuropneumonia are estimated to be two billion ¥ (284 million US$, 258 million €) calculated from the years 2007 to 2017 (personal communication with Prof. Guoqing Shao and Prof. Zhou Li, Jiangsu Academy of Agriculture Sciences, Nanjing, China). *A. pleuropneumoniae* is one of the most prominent RTX toxin-producing bacterial species. The discovery of RTX toxins in *A. pleuropneumoniae* made a significant contribution to the development of novel diagnostic approaches and preventive products to improve swine health (Table 1). The species is currently subtyped into 18 serotypes [72,73]. It contains the following four different RTX toxins: the strongly hemolytic ApxI, the weakly hemolytic ApxII, the non-hemolytic ApxIII, and the “clip and link” protein ApxIV. The ApxIV toxin is produced by all *A. pleuropneumoniae* strains in combination with one or two of the other toxins depending on the respective serotype, where the presence of ApxI, ApxII or ApxIII largely determine virulence. The highly pathogenic strains of serotypes 1, 5, 9, and 11 that cause high mortality and severe outbreaks additionally secrete ApxI and ApxII. Strains of serotypes mostly causing lower mortality but high morbidity, secrete ApxII and ApxIII, while strains with generally low epidemiological impact secrete mostly ApxII [74], in addition to ApxIV. Toxins ApxI, ApxII, and ApxIII are classical RTX toxins with molecular masses of 110 to 120 kDa. ApxI and ApxIII are encoded on classical p*CABD* operons with their proper T1SS (Figure 1). Interestingly, the *apxII* operon in *A. pleuropneumoniae* strains lacks the *BD* genes. ApxII supposedly uses the T1SS encoded by *apxIBD* which was shown to be crucial for virulence [75,76]. Interestingly, the ”missing” genes *apxIIBD* have been found on a full *apxIICABD* operon in *Actinobacillus porcitonsillarum*, a non-pathogenic *Actinobacillus* that is currently not yet officially recognized as a species [77]. This finding, together with small residual 5’ *apxIIB* fragments found on the *apxIICA* loci in most *A. pleuropneumoniae*, indicates that *apxIIBD* genes in *A. pleuropneumoniae* were most likely deleted due to the presence of the very similar *apxIBD*. This most likely occurred since the *apxIBD*-encoded T1SS is able to secrete both ApxI and AxpII [75,76,77,78]. Surprisingly, TolC which also belongs to T1SS, is not annotated on any of the *A. pleuropneumoniae* genomes available at GenBank/NCBI. Full genome sequence analysis of other *Actinobacillus* species have revealed TolC analogs. However, no significant similarity to any of these TolC-like outer membrane channel proteins, including GtxE of *Gallibacterium anatis*, is found in *A. pleuropneumoniae*.

The direct impact of RTX toxins, in particular, ApxI, ApxII, and ApxIII, on virulence was proven by experimental infection of pigs using genetically defined mutants and trans-complemented mutant strains of *A. pleuropneumoniae*. A nonhemolytic serotype 5 mutant lacking the complete *apxICABD* operon was shown to be unable to secrete ApxI or ApxII due to the lack of a T1SS. Despite the residual *apxIICA* genes, it was non-pathogenic to pigs, even at significantly higher infectious doses than the LD_50_ of the parent strain [75]. Partial complementation of this deletion mutant with the type I secretion genes *apxIBD*, restoring secretion of ApxII, resulted in a recombinant strain that caused pleuropneumonia at high doses but no mortality, in contrast to the wild type strain. However, trans-complementation of the same deletion mutant with the entire *apxICABD* operon restored production and secretion of both ApxI and ApxII and resulted in a strain with equivalent virulence to that of the wild type [75]. Similar results were obtained with mutants of a serotype 1 where ApxI^−^/ApxII^−^ mutants were non-virulent in pigs and isogenic mutants ApxI^−^/ApxII^+^ or ApxI^+^/ApxII^−^ phenotype had a reduced virulence [79,80]. Endobronchial inoculation of purified native or recombinant ApxI, ApxII, and ApxIII showed that ApxI, ApxIII, and to a lesser extent ApxII can induce fibrino-hemorrhagic pneumonia and purulent pneumonia. This study revealed the direct role of Apx toxins in triggering the development of clinical signs and lesions typical of porcine pleuropneumonia. The effect of Apx toxins has been attributed to their cytotoxicity for porcine leukocytes, neutrophils, and phagocytes, while the effect on erythrocytes seems to be irrelevant in terms of pathogenesis [81,82,83]. Vanden Bergh and collaborators showed that ApxIII was specifically cytotoxic for peripheral blood mononucleated cells (PMBC) of domestic pigs and wild boars, and did not affect PMBCs of other mammals, including humans, llama, cattle, dogs, rats, mice, and goats [84]. This explains the fact that experimental infections of laboratory mice by virulent *A. pleuropneumoniae* strains did not result in the clinical picture of pleuropneumonia. Furthermore, the authors studied the receptors for ApxIII and showed that a human erythro-leukemic cell line that is deficient in β_2_-integrin is resistant to ApxIII. This cell line can be rendered susceptible to ApxIII by transfection with genes encoding homologous or heterologous CD11a/CD18 heterodimers requiring the CD18 moiety but not necessarily the CD11a to be of porcine origin. Transfectants of these human cells with other heterodimers containing either porcine, human or bovine CD11a, and human or bovine, but not porcine CD18, remained insensitive to ApxIII. This confirmed that the host-specific CD18 moiety is necessary to mediate *A. pleuropneumoniae* ApxIII-induced leucolysis [85]. While the hemolytic potency of the three different major Apx toxins, ApxI, ApxII, and ApxIII, can be explained by their pore-forming activity and the pore sizes generated in artificial lipid bilayers [86], the differences in pathology observed in lung tissue is suspected to be due to other, as yet unknown, attributes of these RTX toxins.

ApxIV is encoded by all *A. pleuropneumoniae* serotypes on a bicistronic operon *orf1 (apxIVC)–apxIVA* as a 200 to 300 kDa RTX toxin, depending on serotype and strain. ApxIV belongs to the calcium-dependent processing and cross-linking “clip and link” proteins characterized by the *Neisseria meningitides* RTX protein FprC [87,88]. ApxIV is not expressed in vitro when grown in axenic cultures in media supplemented with or depleted of various metal ions or growth factors, or under a large variety of culture conditions [88]. In contrast, ApxI, ApxII, and ApxIII are strongly expressed and secreted by the respective strains under standard growth conditions. Furthermore, ApxI production is strongly enhanced by Ca^2+^-enriched medium. ApxIV, however, induces high levels of serum antibodies during infection showing that its operon with the gene *apxIV* is specifically induced upon infection of pigs [64,88,89]. The *apxIV* gene is specific to the species *A. pleuropneumoniae,* and therefore it is a valuable target for PCR detection of *A. pleuropneumoniae* infections in pigs and *A. pleuropneumoniae* contaminations in pig farms, independent of serotype [66,72,90,91,92,93,94]. Since ApxIV is specifically expressed upon infection of pigs and causes seroconversion, recombinant ApxIV was developed as a valuable tool for serological detection of *A. pleuropneumoniae* infections in pigs [64,95]. Because ApxIV is not expressed by *A. pleuropneumoniae* grown in axenic cultures, and therefore is not present in commercialized vaccines that are based on bactrins, culture supernatants, or *A. pleuropneumoniae* subunits, recombinant ApxIV serology can be used to differentiate infected pigs from vaccinated pigs by the DIVA (differentiating infected from vaccinated animals) principle [96]. These features finally led to a broadly used commercial product, IDEXX APP-ApxIV Ab Test® (IDEXX Laboratories Inc. Westbrook, ME, USA), for serodiagnosis of *A. pleuropneumoniae* in pigs and surveillance of *A. pleuropneumoniae* free herds (Table 1).

The discovery of ApxI, ApxII, and ApxIII in the various serotypes of *A. pleuropneumoniae* led to the development of novel vaccines and is still the focus of development for the next generation of vaccines. Vaccines containing toxoided preparations containing all three ApxI, ApxII, and ApxIII showed efficacy against challenge with homologous and heterologous *A. pleuropneumoniae* serovars, constituting an important advance to control porcine pleuropneumonia. A key study showing the impact of RTX toxins in protective immunity in swine was done with the deletion mutant and its trans-complemented derivatives of an *A. pleuropneumoniae* serotype 5 strain described above. While detoxified supernatants from a deletion mutant that cannot secrete either ApxI or ApxII were unable to protect swine against infection with serotype 5 (producing ApxI and ApxII) or serotype 7 (producing ApxII), the trans-complemented mutant that only secreted ApxII protected against serotype 7 but not serotype 5. Only the trans-complemented mutant that secreted ApxI + ApxII protected against both serotypes 5 and 7 (T. Inzana, Greenvale, NY, personal communication). Subunit vaccines based on toxins are among the most advanced and promising developments in RTX-based vaccines. They resulted in successful commercially available vaccines containing three inactivated exotoxins (ApxI, ApxII, and ApxIII). Prominent examples are Porcilis APP_®_, which contains toxoided ApxI, ApxII, and ApxIII and an outer membrane protein (MSD Animal Health) and Coglapix_®_ (Ceva), which contains inactivated *A. pleuropneumoniae* enriched with toxoided ApxI, and ApxII. Further vaccines based on bactrins of various serotypes such as Ingelvac_®_APPX (Boehringer Ingelheim) also incorporate all three antigens, ApxI, ApxII, and ApxIII.

### 2.3. ApxI and ApxII Toxins in Actinobacillus Suis and Actinobacillus Porcitonsillarum

Although RTX toxins ApxI and ApxII are considered as major virulence factors of the primary pathogen *A. pleuropneumoniae*, they are also secreted by other swine *Actinobacillus* species that show low- or non-pathogenic potential. Typically, “*A. porcitonsillarum*”, a non-pathogenic bacterium that is frequently found in the tonsils of healthy pigs and is important in the differential diagnosis of potential carriers of *A. pleuropneumoniae*, actively secretes ApxII via its endogenous T1SS encoded on *apxIICABD* but does not induce lesions in pigs [77,97]. 

Furthermore, *Actinobacillus suis* and (*Actinobacillus) rossii*, are two minor pig pathogens that cause moderate respiratory distress in early-weaned pigs and occasional abortions but not pleuropneumonia. Both secrete significant amounts of ApxI and ApxII (also named AshI and AshII in *A. suis*) [90,98,99,100,101]. This reveals that the significant tissue damage and pathologies described in pigs inoculated tracheally with recombinant RTX toxins, ApxI, ApxII, and ApxIII, which resulted in porcine pleuropneumonia, are not sufficient to produce the disease under natural conditions of infection. *A. pleuropneumoniae* requires further virulence factors such as capsule, iron-uptake-specific LPS and most likely the synergy of the calcium-dependent processing and cross-linking “clip and link” ApxIV toxin to cause disease.

### 2.4. Aqx Toxin from Equine Actinobacillus Equuli

*Actinobacillus equuli* subs. *haemolyticus* is the etiological agent of septicemia in neonatal foals also named “sleepy foal disease”, an illness that is spread worldwide [93,102,103,104]. The hemolytic substance secreted by *A. equuli* subs. *haemolyticus* is a 110 kDa RTX toxin named Aqx, which is encoded on a classic four gene RTX operon *aqxCABD* providing activator, structural toxin, and the T1SS (Figure 1) [67]. The Aqx toxin is considered as the main virulence factor of *A. equuli* subs. *haemolyticus*. It shows 61% amino-acid similarity to the *M. haemolytica* leukotoxin LktA, the phylogenetically most closely related RTX toxin [105]. The Aqx toxin has also been found in the lungs of a mare that exhibited airway-associated hemorrhage in conjunction with a rare bacterial bronchopneumonia caused by *A. equuli* subs. *haemolyticus* [106]. The leukotoxic activity of Aqx has been shown to be specific to equine lymphocytes, in contrast to its hemolytic activity that is equally pronounced on erythrocytes of various animal species including pigs, horses, and sheep [107]. The toxicity of Aqx secreted by *A. equuli* subs. *haemolyticus* is specific to horse leukocytes, while ApxI and ApxII secreted by *A. suis* (and *A. pleuropneumoniae*) target swine leukocytes. This observation provided further corroboration for the impact of RTX toxins in the species specificity of their respective bacterial pathogens. Furthermore, the detection of the specific Aqx toxin and its genes *aqxCABD* in *A. equuli* provided evidence for the taxonomic differentiation of *A. equuli* subs. *haemolyticus* from *A. suis* (Table 1). These two species were often confounded, in the past, due to their high phenotypic resemblance. Hence, the discovery and characterization of the Aqx toxin provided a significant improvement in bacterial diagnostics of equine diseases using specific RTX gene-based PCR assays.

While “sleepy foal disease” is an infectious disease, it generally affects only single foals and shows no outbreaks even though *A. equuli* subs. *haemolyticus* is very frequently found among healthy mares. An epidemiologic study of anti-Aqx antibodies in healthy mares and foals revealed that adult horses and foals generally exhibited neutralizing anti-Aqx antibodies [108,109], explaining earlier observations of the neutralizing activity of sera of mares towards the hemolysin of *A. equuli* subs. *haemolyticus* reported by Rycroft and collaborators [110]. In contrast, foals are seronegative for Aqx directly after birth and acquire neutralizing anti-Aqx antibodies within one hour of colostrum intake in a time-dependent manner. Adult horses which are generally carriers of *A. equuli* subs. *Haemolyticus,* hence, possess anti-Aqx antibodies that neutralize the toxic activity of Aqx while newborn foals need to acquire it via the maternal colostrum. Therefore, it is assumed that the pathologies of “sleepy foal disease” caused by *A. equuli* subs. *haemolyticus* are due to a failure in acquisition of neutralizing antibodies from a mare’s colostrum [109]. This also explains the non-epidemic incidence of the disease.

### 2.5. GtxA MARTX from the Poultry Pathogen Gallibacterium Anatis

*Gallibacterium anatis* has emerged in the last few years as an important multi-antibiotic resistant pathogen in intensively reared poultry birds causing losses in production. It causes heavy mortality in broiler chickens and a drop in egg production in layer hens by triggering salpingitis, peritonitis, salpingoperitonitis, bacteremia, oophoritis, follicle degeneration, hepatitis, enteritis, and respiratory tract diseases [111,112,113,114]. *G. anatis* secretes a strongly cytotoxic and hemolytic RTX toxin named GtxA, which has a particularly high molecular mass of 215 kDa and an uncommon domain structure [115]. A Δ*gtxA* knock-out-mutant demonstrated that GtxA is responsible for both the hemolytic and the leukotoxic activity of *G. anatis*. GtxA has an uncommon modular structure resembling that of multifunctional autoprocessing repeats-in-toxin (MARTX) toxin such as the adenylate cyclase toxin, CyaA, from *Bordetella pertussis* [116]. The C-terminal half of GtxA is analogous to classical pore-forming RTX toxins and includes the amphiphatic helical domain, the strong hydrophobic domain, the glycine-rich nonapeptide repeats, and two conserved lysine residues as potential acylation sites. The N-terminal domain is different from RTX proteins but is necessary for the cytotoxic activity of GtxA [117]. This part of GtxA shows some sequence similarity to the eukaryotic cytoskeletal proteins Talin-A/B from the amoeba *Dictyostelium discoideum,* and hence could have the capacity to bind to actin, vinculin, and integrins [117]. It is interesting to note that the N-terminal domain of GtxA shows significant amino-acid similarity (46% identity and 72% similarity) to a yet unassigned hypothetical protein of *Suttonella ornithocola* (JF personal data). *S. ornithocola* is a bacterial species that was recently isolated from carcasses of tits (*Paridae*) in Great Britain and Germany [118,119]. *S. ornithocola* is suspected to be a fowl pathogen causing necrotic pneumonia in coal tits (*Periparus ater*), blue tits (*Cyanistes caeruleus*), and great tits (*Parus major*) [118].

The genetic structure of *Gallibacterium anatis* RTX toxin consists of two separate operons. One contains the structural toxin protein, *gtxA,* followed by the activator gene, *gtxC*, surprisingly in reverse orientation as compared with classical RTX operons. The other is a tricistronic operon encoding the T1SS by the genes *gtxEBD*, where *gtxB* and *gtxD* are *rtxBD* analogs and *gtxE* represents the *tolC* analog that is found in most other bacterial species on a separate chromosomal locus [115] (Figure 1). The role of *gtxC* as the activator gene of GtxA has been confirmed by heterologous expression of the *gtxAC* genes that produce cytotoxic GtxA or *gtxA* alone that produces the non-toxic preprotein. GtxA is the dominant hemolysin of *Gallibacterium* species and the *gtxAC* genes are present in strongly hemolytic, as well as in weakly hemolytic and non-hemolytic *G. anatis* strains. It was shown that differential expression of the *gtxCA* genes in various strains of *G. anatis* was the cause of the significant differences in hemolysis and cytotoxic activity of these strains [115]. Vaccination experiments using either outer membrane vesicles and fimbrial protein FlfA or recombinant Gtx-N’ (N-terminal half of GtxA) and Gtx-C’ (C-terminal half of GtxA) together with FlfA resulted in protection against challenge with heterologous strains of *G. anatis*. This demonstrates the potential of GtxA in the development of novel vaccines against *G. anatis* infections in poultry (Table 1) [111,120,121,122].

### 2.6. AvxA MARTX from Poultry Pathogen Avibacterium Paragallinarum

*Avibacterium* (*Haemophilus*) *paragallinarum* is the etiological agent of infectious coryza, a severe acute respiratory disease of chickens that affects mainly laying hens and causes considerable economic losses in egg production [123,124]. Genes, for a classical RTX operon *avxCABD,* were initially detected in strain H18 and further demonstrated by PCR in a large collection of *A. paragallinarum* strains. The *avxA* gene was shown to encode a large, novel structural RTX toxin-like protein with a predicted molecular mass of about 250 kDa containing a peptidase S8 domain and a proprotein convertase P-domain [125]. Concurrently, *A. paragallinarum* type strain ATCC29545 and field strains of the major serovars A, B, and C were found to secrete a heat-labile cytotoxic protein that was identified as an RTX toxin AvxA belonging to the multifunctional autoprocessing repeats-in-toxin (MARTX) subfamily [68]. The toxin is encoded on a classical polycistronic RTX operon structure with the activator gene *avxC*, the structural bivalent serine-protease-RTX toxin gene *avxA*, and the T1SS genes *avxBD* encoding a proper type I secretion system (Figure 1). The cytotoxic activity of AvxA was shown to be activated by the *avxC* gene using the heterologous expression of *avxCA* genes in *E. coli* that resulted in cytotoxicity while the expression of *avxA* alone was non-cytotoxic. AvxA contains an N-terminal serine-protease moiety and a C-terminal RTX moiety. AvxA is proteolytically processed leaving a 95 kDa RTX porin moiety that is found in culture supernatants of *A. paragallinarum* serovars A, B, and C. This RTX moiety of AvxA (AvxA-RTX) shows significant similarities (57% identical and 74% similar amino acids) with the RTX segment (C-terminal half) of the GtxA toxin of *G. anatis*. AvxA-RTX shows cytotoxic specificity against avian macrophages. It is cytotoxic to the macrophage-like cell line HD11 but not to the bovine macrophage cell line BoMac [68]. Purified IgG from hyperimmune rabbit anti-AvxA-RTX serum, made by immunization of rabbits with recombinant AvxA-RTX from a serotype A strain, fully neutralizes the cytotoxic activity of recombinant active AvxA-RTX and toxic culture supernatants of *A. paragallinarum* serotypes A, B, and C. Furthermore, sera collected from chickens exposed to *A. paragallinarum* were found to exhibit strong reactivity to the AvxA protein, suggesting that AvxA is immunogenic [125]. These findings are still preliminary but indicate that AvxA is a common major virulence attribute of all *A.paragallinarum* serotypes and represents a promising target for future subunit or recombinant vaccine developments against infectious coryza (Table 1).

## 3. Conclusions

RTX and MARTX toxins are widely spread among animal Gram-negative bacterial pathogens. They are thought to have originated from *Pasteurellaceae*, from where they have spread to other bacterial families [126]. RTX toxins or genes encoding RTX proteins have also been found in the animal pathogens *Moraxella bovis*, pathogenic serovars of *Escherichia coli*, *Bordetella bronchiseptica*, *Pasteurella aerogenes*, *Pasteurella trehalosi*, *Pasteurella mairi*, *Mannheimia glucosida*, and *Mannheimia varigena*. New developments in full genome sequencing techniques could facilitate the discovery of new RTX toxins in pathogens that have not yet been analyzed at the molecular level. Research has mainly been done on RTX toxins from the most relevant animal pathogens. These studies contributed to new fundamental knowledge, particularly describing mechanisms of toxin–receptor interaction and the role of RTX toxins in host specificity. This characteristic explains the difficulty to reproduce the appropriate diseases of RTX-producing pathogens in laboratory animals. New knowledge of the high specificity of receptors of RTX toxins in the susceptible hosts opens new ways to improve animal breeding and selection of disease resistant breeds. In addition, much research effort has been made investigating RTX toxins and their genes from animal pathogens, focusing on the development of novel methods for diagnostics with improved specificity. These findings have been beneficial for molecular epidemiology and control of major epizootic diseases. The capacity of RTX toxins to induce a strong adaptive immunological response, including neutralizing antibodies, has led to a better understanding of the course of infectious cycles and the development of efficient vaccines to prevent epizootics in farmed animals. The development of DIVA approaches in vaccine design could have a particular impact by overcoming the issue of distinguishing vaccinated animals from infected animals. While this is currently possible for certain *A.pleuropneumoniae* vaccines, solid validation data are necessary for the approval of functional DIVA vaccines for most other applications in practice. While modern genetic methods allow rapid detection of novel RTX toxins in pathogens, functional and detailed biochemical and immunological studies on their role in pathogenicity is necessary to be able to use the knowledge of RTX toxins in translational medicine.

## Figures and Tables

**Figure 1 toxins-11-00719-f001:**
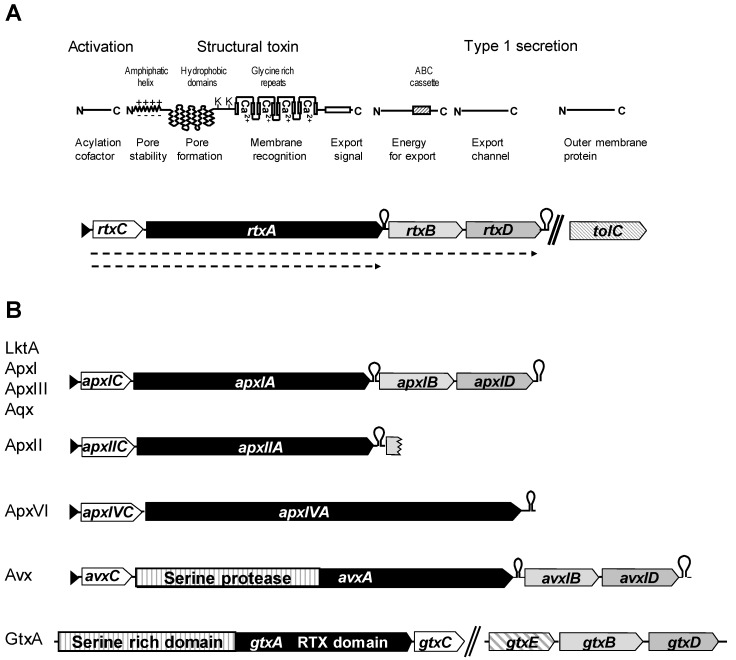
Genetic organization and functional domains of repeats in the structural toxin (RTX) operons. (**A**) Represents the common genetic and structural organization of RTX toxins and is mainly based on that of the *E. coli* hemolysin HlyA. The basic functional activities are given on the top line, followed by a schematic statement of the gene products and their major structural characteristics. These domains are annotated. K indicates the lysine acylation sites. The ATP-binding cassette of the secretion protein B is abbreviated by ABC. N indicates the amino-terminal end and C the carboxy-terminal end of the peptides. The four genes of the operon and the unlinked outer membrane gene *tolC* are represented by arrowhead boxes, which indicate the relative length and direction of the coding genes. Black triangles represent transcription promoters and the hairpins show the sites of rho-independent transcription termination signals. The dashed arrows on the bottom of (A) represent the direction and length of transcripts as determined in *hlyCABD* of *E. coli* and *apxCABD* of *A. pleuropneumoniae* [11,12]. (**B**) The genetic organization of the other types of RTX and multifunctional autoprocessing repeats-in-toxin (MARTX) operons as indicated on the left side. Shading of the different arrowhead boxes was done in analogy to the prototype RTX toxin and *tolC* genes of (A). The non-RTX domains of the MARTX toxins are dashed.

**Table 1 toxins-11-00719-t001:** RTX toxins and their respective genes from prominent animal pathogens that were used for diagnostic or therapeutic applications.

RTX Toxin *rtx* Gene	Species	Use in Veterinary Medicine
ApxI	*A. pleuropneumoniae*	Antigen in commercial, universal serovar vaccines against porcine pleuropneumonia [63]
ApxII	*A. pleuropneumoniae*	Antigen in commercial, universal serovar vaccines against porcine pleuropneumonia [63]
ApxIII	*A. pleuropneumoniae*	Antigen in commercial, universal serovar vaccines against porcine pleuropneumonia [63]
ApxIV	*A. pleuropneumoniae*	Recombinant ApxIV ELISA for sero-detection of *A. pleuropneumoniae* infected pigs [64]
*apxI*	*A. pleuropneumoniae*	Diagnostic PCR for toxin typing of *A. pleuropneumoniae* strains [65]
*apxII*	*A. pleuropneumoniae*	Diagnostic PCR for toxin typing of *A. pleuropneumoniae* strains [65]
*apxIII*	*A. pleuropneumoniae*	Diagnostic PCR for toxin typing of *A. pleuropneumoniae* strains [65]
*apxIV*	*A. pleuropneumoniae*	Diagnostic PCR detection of *A.pleuropneumoniae* [66]
*aqxA*	*A. equli*	Diagnostic PCR for identification of *A. equuli* subsp. *haemolyticus* [67]
AvxA	*A. paragallinarium*	Recombinant AvxA-RTX for development of vaccines against infectious coryza [68]
*avxA*	*A. paragallinarium*	Diagnostic PCR for *A. paragallinarium* species confirmation [68]
GtxA	*G. anatis*	Antigen in experimental vaccines against *G. anatis* infections in layer hens [69]
LktA	*M. haemolytica*	Antigen in commercial vaccines against mannheimiosis and *B. threalosi* infections [61,62]
*lktD*	*M. haemolytica*	Diagnostic multiple PCR for bovine respiratory disease complex (BRDC) [70]
*lktA*	*M. haemolytica*	Diagnostic multiple PCR for bovine respiratory disease complex (BRDC) [71]

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
