# Peer review of "RTX Toxins of Animal Pathogens and Their Role as Antigens in Vaccines and Diagnostics"

_toxins, 2019, doi:10.3390/toxins11120719_

Round 1

Reviewer 1 Report

The manuscript appears well written and detailed in the description of molecular mechanisms adopted by RTX for exerting their pathogenicity. For these reasons it is worthy of publication in Toxin. In addition, the literature lacks of papers reviewing this topic.

In my opinion the manuscript could be improved including some sentences regarding the burden of the different diseases described in the paper. These information will help reader to understand the importance and the impact of these diseases on animal population.

Finally, in the Conclusions section Authors should include some sentences detailing the impact of research efforts on RTX toxins, for example overcoming the gaps on diagnotic methods and in vaccine production, also considering the new strategies for candidate antigens identification and vaccine production. 

Author Response

I have added new data on the estimation of the economic burden of procine pleuropneumonia caused by Actinobacillus pleuropneumoniae in th ebiggest pig producing coutnry China. Data from other countries that are often cited in papers are outdated as they were extimates from the 1990ies. No reliable data are available from other countries or about other diseases caused by RTX producing pathogens.

Impact of research effort on RTX toxins in the animal health domain have been added to the conclusions section.

The manuscript was revised by a professional scientific proof reading service

Reviewer 2 Report

Authors review RTX toxins of bacterial animal pathogens, including regulatory operons and biological activities. The article is not easy to follow the interpretation, which lacks of the attractive figures of structure-activity relationship and pathogenic pathways, as well as tables of classifying toxins and markers for diagnostics and therapeutics.

Author Response

The paper has been significantly revised for better understanding and was revised for English by a professional scientific proof reading service.

Table 1 has been added for classifying toxins and toxin genes for developments in diagnsotics and therapeutics.

Reviewer 3 Report

The review is informative and generally well written. However, there are lots of typos (some of them are listed below) throughout the manuscript that need to be fixed before publication.

line 59: Change '… RTX toxins is characteristic and' to '… RTX toxins is characterized that…'

line 64: Change ' Gene tolC that is also required but not exclusive for the T1SS is generally not part of the operon and is located at a distant site on the bacterial chromosome.' to 'Gene tolC, that is also required but not exclusive for the T1SS, is generally not part of the operon and is located at a distant site on the bacterial chromosome.'

Change the font type to italic:

line 151-152: M. haemolytica, Pasteurellaceae, B. trehalosi

line 178: A. pleuropneumoniae

Please correct the grammar of these sentences:

line 155-156: Actinobacillius pleuropneumoniae, the etiological agent of porcine pleuopneumonia a severe highly contagious acute or chronic infections disease

line 262: …[Actinobacillus] rossii two minor pig pathogens that cause moderate respiratory distress in early weaned pigs respectively occasional abortions, both secrete significant amounts of ApxI and ApxII

line 182: Change 'ApxI ApxII and ApxIII' to 'ApxI, ApxII, and ApxIII'

line 330: Change '…analogue That is found…' to '…analogue that is found…'

line 338: Please clarify the meaning of GtxN' and GtxC', are they the N-terminal domain and C-terminal domain of GtxA, respectively?

Author Response

The suggestions for corrections made by reviewer 3 have been introduced to the revised manuscript.

In addition the manuscript has been revised profoundly for English by a professional scientific proof reading service.

Round 2

Reviewer 2 Report

Authors had responed the comments.